# Exploiting Partial Solubility in Partially Fluorinated Thermoplastic Blends to Improve Adhesion during Fused Deposition Modeling

**DOI:** 10.3390/ma15228062

**Published:** 2022-11-15

**Authors:** Pau Saldaña-Baqué, Jared W. Strutton, Rahul Shankar, Sarah E. Morgan, Jena M. McCollum

**Affiliations:** 1Department of Mechanical and Aerospace Engineering, University of Colorado Colorado Springs, Colorado Springs, CO 80918, USA; 2School of Polymer Science and Engineering, University of Southern Mississippi, Hattiesburg, MS 39406, USA

**Keywords:** adhesion, thermoplastic, crystallinity, fused deposition modeling, injection molding, tensile testing

## Abstract

This work studies the effect of interlayer adhesion on mechanical performance of fluorinated thermoplastics produced by fused deposition modeling (FDM). Here, we study the anisotropic mechanical response of 3D-printed binary blends of poly (vinylidene fluoride) (PVDF) and poly (methyl methacrylate) (PMMA) with the isotropic mechanical response of these blends fabricated via injection molding. Various PVDF/PMMA filament compositions were produced by twin-screw extrusion and, subsequently, injection-molded or 3D printed into dog-bone shapes. Specimen mechanical and thermal properties were evaluated by mode I tensile testing and differential scanning calorimetry, respectively. Results show that higher PMMA concentration not only improved the tensile strength and decreased ductility but reduced PVDF crystallization. As expected, injection-molded samples revealed better mechanical properties compared to 3D printed specimens. Interestingly, 3D printed blends with lower PMMA content demonstrated better diffusion (adhesion) across interfaces than those with a higher amount of PMMA. The present study provides new findings that may be used to tune mechanical response in 3D printed fluorinated thermoplastics, particularly for energy applications.

## 1. Introduction

Additive manufacturing (AM) allows the production of complex geometries that are oftentimes not achievable by subtractive methods [1]. In the particular case of fused deposition modeling (FDM), deep understanding of adhesion phenomena that occur during thermoplastic printing remains an open problem. Adhesion not only plays a vital role in AM techniques but also in many aspects of most manufacturing methods [2]. In extrusion-based printing methods, materials must be designed to enhance interfacial adhesion during the printing process to minimize anisotropic static and dynamic properties caused by microstructural heterogeneity [3]. Polymer chains must reach a high enough temperature (i.e., above the glass-transition temperature (*T_g_*)) so that they are mobile and can diffuse through interlayer boundaries. Ideally, the mechanical response of a printed thermoplastic (multilayered specimen) would be similar to the isotropic mechanical response of the same thermoplastic fabricated via molding techniques (homogeneous specimen). By performing mechanical testing on these specimens, one can probe interface diffusion by observing behavior in the bulk [4,5].

The FDM process consists of a heated print nozzle through which a polymer filament passes undergoing melting. The extruded filament is then deposited onto a build platform in a chosen xy-plane, while building layer-upon-layer in the vertical z-dimension, allowing the production of complex geometries (see Figure 1). In this process, the adhesion quality among thermoplastic interfaces depends on polymer structure [6], molecular weight [7], convective and conductive exposure [8], blend composition [9,10], and printing parameters [11] (e.g., speed, temperature, etc.). Quantifying adhesion in this context is of utmost importance because it will determine the integrity and mechanical properties of the manufactured materials [3].

Polymer surfaces brought into contact must be mutually soluble and above their glass-transition temperature (*T_g_*) or melting temperature (*T_m_*) to achieve sufficient molecular mobility such that polymer chains can diffuse and entangle with one another across the interface. The polymer diffusion and entanglement formation determine much of the adhesion strength. In an optimum adhesion between surfaces, a failure will occur cohesively in the specimen rather than at the interface between two layers. There are many factors involved in the diffusion and entanglement mechanism, such as, temperature, contact time, molecular weight, degree of crystallinity, the strength of polar groups [12]. The main objective of this study is to understand and probe interface diffusion to know the factors that affect layer adhesion in partially miscible, fluorinated thermoplastic blends (i.e., poly (vinylidene fluoride) (PVDF) and poly (methyl methacrylate) (PMMA)).

PVDF is a fluorinated semicrystalline thermoplastic with a glass-transition temperature of approximately −40 °C and a melting point around 170 °C [13]. Depending on its conformation, PVDF can crystalize into five different crystalline phases (polymorphs): α, β, γ, δ and ε [14,15]. The first three phases (α, β,) are the most commonly studied, with the α-phase being the most thermodynamically preferred [16]. When PVDF crystallizes into β and/or γ phases, dipole stacking induces piezoelectric properties, which enables its use as a sensor [17] or actuator [18]. For use in FDM, PVDF alone has very low surface energy, which can make 3D printing difficult. One strategy to mitigate low surface energy is to incorporate a miscible, high surface energy polymer (i.e., PMMA). Owing to its high surface energy and partial miscibility with PVDF, PMMA plays an essential role by increasing the overall surface energy of the blend [19,20,21], which enables improved production of energetic thermoplastics through FDM [20]. Furthermore, PVDF and PMMA blends are of interest due to their potential as an oxidizing binder for energetic materials [20], their potential in sensing and actuating systems [20], and their use as a separator in lithium-ion batteries due to high polarity and electrolyte uptake [22,23,24,25]. The key objective of this study is to assess mechanical durability of PVDF/PMMA blends that have been manufactured by FDM alongside injection molding to tease out the role of adhesion on mechanical durability in partially fluorinated polymer blends.

## 2. Materials and Methods

### 2.1. Materials

Poly (vinylidene fluoride) (PVDF) (Kynar 705) was generously supplied by Arkema (King of Prussia, PA, USA) and arrived in pellet form. Poly (methyl methacrylate) (PMMA) with an average molecular weight of 141,000 g/mol was purchased from Sigma Aldrich (St. Louis, MO, USA) and arrived in powder form. “PMMAXX” is the nomenclature structure for samples used in this study where “XX” indicates weight-based PMMA concentration. For instance, a sample made with 25 wt% PMMA and 75 wt% PVDF is designated as “PMMA25”. All materials used in this study were obtained from commercial sources and dried at 50 °C under vacuum for 8 h to remove any moisture before use.

### 2.2. Blend Preparation

Blends were prepared by melt blending PVDF and PMMA in a twin-screw compounder (HAAKE Minilab II, Thermo Scientific, Waltham, MA, USA). Compounding was carried out at 190 °C and a screw speed of 100 rpm for 5 min. Different PVDF/PMMA blends were extruded at 100 rpm to form filament approximately 3 mm in diameter, containing 0%, 15%, 25%, 50%, 75%, 85%, and 100% of PMMA (by weight).

### 2.3. Dog-Bone Production

HAAKE Type 3 specimen (557-2290) dog-bone tensile bars were produced via FDM and injection molding. All samples of each composition were fabricated under identical processing conditions to eliminate sample-to-sample errors. Examples of dog bones produced by FDM and injection molding are shown in Figure 2.

#### 2.3.1. Fused Deposition Modeling (FDM)

PVDF/PMMA melt-extruded filaments were around 0.5–0.7 m in length and 2.9–3.1 mm in diameter. These filaments were used in an Ultimaker 2 to create the 3D printed dog-bones. Solidworks 2019 (Dassault Systemes, Vélizy-Villacoublay, France) and Simplify3D (Simplify3D, Cincinnati, OH, USA) were used to model the dog-bones prior to the printing process. Printing of the specimens was done on a glass build plate heated to 90 °C through a 0.4 mm nozzle heated to 190–210 °C. For blends from PMMA0 to PMMA25, the print nozzle was heated to 190 °C. For blends from PMMA50 to PMMA100, the print nozzle was heated to 210 °C. The layers were printed with 100% infill density, with an extrusion multiplier of 1, with high quality, with a layer height of 0.1 mm, at a print speed of 30 mm/s, and a x/y movement speed of 55 mm/s. 

#### 2.3.2. Injection Molding (IM)

PVDF/PMMA melt-extruded filaments were pelletized and introduced into a preheated cylinder (210 °C). A HAAKE Minijet Pro (Thermo Scientific, Waltham, MA, USA) was then used to form the dog-bone via injection molding. The cylinder was placed on top of a heated mold (90 °C) and fitted with a plunger. The plunger was pushed by a hydraulic press to extrude the molten blend into the closed mold. For blends from PMMA0 to PMMA25, the press was programmed to deliver an initial injection pressure of 420 bar for 20 s, and a secondary pressure of 200 bar for 15 s before releasing. For blends from PMMA50 to PMMA100, the press was programmed to deliver an initial pressure of 600 bar for 40 s, and a secondary pressure of 250 bar for 20 s before releasing. Upon completion, the mold was then removed from the machine and opened to eject the injected dog-bone.

When injection molding PMMA (or blends with high wt% in PMMA), there exists a wide range of adjustable injection temperatures (usually between 180 to 240 °C) since its decomposition temperature is approximately 270 °C [26]. Furthermore, this material is hygroscopic and has high viscosity making it difficult to injection mold. As a result, during this process, high temperature and injection pressure are required. It is important to note that, although the improvement of fluidity through injection temperature is greater, the increase in injection pressure helps to decrease specimen shrinkage. Regarding the mold, its temperature should be raised (e.g., approximately 60 to 90 °C) to improve the solidification process, since PMMA has poor impact and abrasion resistance. Simultaneously, the mold temperature may also significantly influence PVDF crystallization, either positively or negatively.

### 2.4. Methods

#### 2.4.1. Apparent Viscosity Measurements

Blends apparent viscosity was measured by monitoring the two pressure transducers located in the inside wall of the compounder chamber. Pressure data were taken at each transducer at 1 Hz for 1 min at a screw speed of 100 rpm using the HAAKE minilab software. Viscosity calculations were made using the pressure difference between the two transducers. Apparent viscosity can be calculated using Equation (1), where η denotes apparent viscosity, $\dot{\gamma }$ represents shear rate, and *τ* is the shear stress at the wall.
(1)$$\eta =\frac{\tau }{\dot{\gamma }}\;$$

The latter is given by Equation (2), where *h* and *w* are the height and width of the channel (i.e., 1.5 and 10 mm, respectively), Δ*L* is the distance between the pressure transducers (i.e., 64 mm), and Δ*P* is the pressure differential.
(2)$$\tau =\frac{hw\Delta P}{2\left(h+w\right)\Delta L}$$

Similarly, the shear rate can be expressed as shown in Equation (3), where *w* represents screw speed.
(3)$$\dot{\gamma }=\frac{2\;\left(h+w\right)w}{60\Delta L}$$

#### 2.4.2. Thermal Analysis

Thermal analysis was carried out using a TA Instruments model Q20 differential scanning calorimeter (DSC) (TA Instruments, New Castle, DE, USA). DSC was performed on raw materials, extruded filament, injection molded samples, and 3D printed samples. PVDF/PMMA samples of approximately 5 mg were placed into an aluminum crucible and loaded into the DSC. Thereupon, these samples were heated at a rate of 20 °C/min, from 30 to 250 °C, and then cooled to 30 °C for two cycles under a nitrogen atmosphere. The glass-transition temperature (*T_g_*), melting temperature (*T_m_*), crystallization temperature (*T_c_*), and heat of melting (Δ*H_m_*) were determined for the different processed blends. To assess the impact of PMMA on PVDF crystalline domains, the degree of crystallinity (*X_c_*) of each sample was calculated from the following Equation (4) [27], where Δ*H_m_* is the experimental melting enthalpy of the blend (obtained through DSC analysis), $\Delta H_{m}^{*}$ represents the theoretical melting enthalpy for 100% crystalline PVDF (i.e., 104.5 J/g [28,29,30]), and ϕ is the weight fraction of PVDF.
(4)$$X_{c}\left(\%\right)=\frac{\Delta H_{m}}{\Delta H_{m}^{*}\phi }100$$

DSC analysis was performed from the derivative of heat flow with respect to temperature. The derivative was used to establish the initial and final points of the exothermic/endothermic peaks. Since crystalline domains affect interface adhesion by preventing the diffusion of polymer chains, DSC was used to locate when crystallization enhancement occurs.

#### 2.4.3. Tensile Testing

Bulk tensile testing of dog-bones was carried out using an Instron 5569 Tensile Tester (Instron, Norwood, MA, USA) with a 50 kN load cell. Dog-bones were placed in a pair of manual dual screw press clamps which were torqued to 40 N·m before testing. The displacement of the sample was measured using the crosshead of the machine frame. Tests were performed at room temperature using a crosshead speed of 50 mm/min, while axial strain was continuously recorded. The Young’s modulus (*E*) was calculated from the initial linear portion of the stress–strain curves. Maximum tensile stress (*σ*) and elongation at break (*ε*) were also reported. At least eight dog-bones specimens were tested for each composition and the average values are presented. Bluehill Universal software was used to measure and calculate stress–strain data.

The main objective is to examine the mechanical response of 3D printed thermoplastics (multiple interface specimens) and of thermoplastics fabricated via injection molding (homogeneous specimens) to study adhesive versus cohesive failure mechanism (i.e., study the adhesion/diffusion quality).

#### 2.4.4. Attenuated Total Reflectance Spectroscopy (ATR-FTIR)

The attenuated total reflectance (ATR) sampling technique was used in conjunction with infrared spectroscopy to determine the presence and relative amount of PVDF β-phase within the blends. Thus, to identify the PVDF crystalline phases of PVDF/PMMA mixtures, absorption spectra were collected using a Thermo Scientific FTIR-ATR Nicolet iS20. The analysis was performed at room temperature in the range of 500 to 3000 cm^−1^ averaged from 32 scans with a spectral resolution of 2 cm^−1^. A background spectrum was collected before each sample scan.

#### 2.4.5. Gel Permeation Chromatography (GPC)

The basic idea behind this chromatographic method is that the polymer in solution passes through columns that absorb and filter it, while a detector measures the size of its molecular chains over time. The average size corresponds to the polymer molecular weight (M_w_). The number of molecules per size corresponds to the molecular weight distribution. It is worth noting that, apart from the aforementioned effect that M_w_ has on diffusion, most polymer properties also depend on the molecular weight and its distribution.

The molecular weight and polydispersity of PMMA were determined via size exclusion chromatography (SEC). GPC was conducted using a Waters Alliance 2695 separations module (Waters Corporation, Milford, CT, USA), an online multiangle laser light scattering (MALLS) detector fitted with a gallium arsenide laser (20 mW) operating at 658 nm, an interferometric refractometer operating at 65 °C, and 685 nm, and two PLgel mixed-E columns in series (pore size 50–103 Å, 5 μm bead size). The mobile phase was freshly distilled tetrahydrofuran (THF) delivered at a flow rate of 1.0 mL/min. Sample concentrations were 6 mg of polymer/mL of THF, and the injection volume was 100 μL. The detector signals were simultaneously recorded using ASTRA software, and absolute molecular weights were determined by MALLS using a *dn/dc* value obtained from the interferometric refractometer response, assuming 100% mass recovery from the columns.

For PVDF, the molecular weight and polydispersity were also determined via the same procedure. However, the mobile phase was 0.02 M LiBr/DMF (dimethylformamide) delivered at a flow rate of 0.5 mL/min. Furthermore, the polymer samples were pre-dissolved in 0.02 M LiBr/DMF by stirring for 3 h at 80 °C. The injection volume was set at 100 μL.

## 3. Results and Discussion

### 3.1. Apparent Viscosity Calculations

As shown in Table 1, a strong rise in apparent viscosity can be observed with increasing blends of PMMA content. This increasing trend in viscosity reflects the high tackiness (M_w_) of PMMA, which causes the melt blending process to consume more energy (i.e., fluid has a higher interaction with the compounder wall relative to shear rate). In contrast, the relatively low PVDF surface energy reduces these interactions and plasticizes PMMA. More precisely, at low PMMA concentrations, the viscoelastic behavior suggests the blends have a high degree of partial miscibility, whereas at PMMA50 a more significant phase separation of the two polymers can be inferred. For higher PMMA concentrations, blends exhibit responses resembling pure PMMA.

These viscosity trends suggest that interface diffusion worsens with the addition of PMMA, since it hinders the mobility of polymer chains. This can be easily understood by looking at the Newtonian polymer-sintering model proposed by Bellehumeur and colleagues [3]. It can be inferred from this model that to achieve good wettability (i.e., polymer chains interact more efficiently because the contact area is larger) thermoplastics should have high surface tension and low viscosity. Although qualitatively the increase in blends apparent viscosity should worsen wetting and diffusion, the increase in global surface energy produced by the addition of PMMA plays a more dominant role for low PMMA concentrations than the increase in viscosity (see next section).

### 3.2. Thermal Analysis

Differential scanning calorimetry (DSC) was performed to study the effect of the addition of PMMA on the thermal behavior of both processed thermoplastic and their respective mixtures. The DSC results of the investigated PVDF/PMMA blends are listed in Table 2 and Appendix A. Moreover, the heat flow curves of the blends are represented in Appendix A. It can be observed from the second heating cycles that as the PMMA con

Centration increases, the melt temperature, the melting enthalpy, and the degree of crystallinity decrease markedly for all processed blends. This behavior is caused by the partial miscibility of PVDF in PMMA, as PMMA is largely amorphous and does not contribute to the heat of fusion [19,29], Thus, the increase in PMMA content disrupts PVDF crystalline domains. More specifically, PMMA chains impede PVDF crystallization due to the high glass-transition temperature (*T_g_*) of PMMA, below which the overall chain mobility significantly decreases by forcing PVDF chains to diffuse through the rigid PMMA network to form crystals, thus altering melting behavior. In this way, crystallization indicates that there is still mobility in the system (i.e., interdiffusion of PMMA chains through amorphous PVDF regions). These results suggest that PMMA50 is the limiting concentration beyond which the PVDF crystallization event is lost.

As mentioned in the previous section, the addition of PMMA raises the glass-transition temperature of the mixtures (see Table 2). Thus, by controlling and maintaining the temperature between the glass transition of PMMA and the melt temperature of PVDF, blends diffusion and crystallization may be optimized [20]. Moreover, pure PVDF exhibits a single endothermic peak around 173 °C, which could be ascribed to the presence of α-phase crystals [19,29]. This was confirmed by the results shown in Appendix A.

The thermal behavior of injection molded blends presents a similar profile with that of 3D printed and compounded (filament). However, the melt temperature of the PMMA15 concentration increases with respect to pure PVDF. This shift could be ascribed to the increase in α-phase crystals of PVDF. Similarly, the degree of crystallinity of the PMMA25 concentration increases with respect to PMMA15, which could be attributed to the slight increase in β-phase content. This was confirmed by the results shown in Appendix A. Furthermore, it can be seen from the second heating of Appendix A that low PMMA injection molded samples experienced significantly less overall crystallinity than the same concentrations of the other processing techniques.

The above reasoning is extrapolated from the second heating-cooling cycle (Appendix A) of the DSC analysis because the first cycle (Table 2) was used as an annealing process to relieve part of the residual stresses induced during production; thus, it is not representative of the blend composition. Differences between the crystallization-melting behavior of the first and second cycles are processing-related phenomena and not composition-related phenomena. In fact, the changes observed between the first and second cycles could simply be a polymer chain readjustment to relieve the internal stresses (i.e., residual stress relaxation) caused during production. Hence, the first DSC cycle shows all the processing effects while the effects in the second cycle are more representative of the material composition. That is, samples act more like virgin material in the second cycle, whereas some of the interactions and phase transitions in the first cycle (and some of the mechanical responses) could simply be a residual stress relaxation issue.

The surface of the material experiences relatively high shear stress during production resulting in internal residual stress. Inside the nozzle, this shear stress aligns polymer chains that are in a low entropy state; they have very few degrees of freedom. Ideally, once the material exits the nozzle, forces acting on it are relieved. However, if the material cools too quickly, the polymer chains do not have enough time to reorder at the surface, resulting in strain-induced crystallization or other types of stress phenomena. This shear stress is more impactful for high PMMA concentrations due to their high glass-transition temperature, which hinders blends readjustment. Moreover, the high surface energy of PMMA increases the tackiness (i.e., adhesive properties) of the system, facilitating interactions between the polymer and nozzle (or mold) surfaces. Therefore, both events demonstrate that higher PMMA concentrations add more shear stress to the material. Similarly, 3D printed components are more likely to experience higher residual stresses and process effects than the other two processes based on the surface area that is formed during production. It is then clear that in the 50% PMMA concentration composition does not promote crystallization. Rather, processing promotes strain-induced crystallization (see Table 2). Lastly, it is important to note that the Flory-Fox Equation for *T_g_* calculation of polymer blends does not apply in this case due to the approx. 160 °C difference between the PVDF and PMMA glass transition temperatures.

### 3.3. Tensile Testing

Mechanical properties as a function of PMMA weight fraction are displayed in Figure 3. 3D printed specimens show a similar trend as injection molded (IM) ones with respect to tensile strength and Young’s modulus for increasing PMMA concentrations. The overall increase in tensile strength by having more PMMA content in the mixture can be explained by dividing the mechanical response into three ranges. The first range is from PMMA0 to PMMA25, the second one is PMMA50, and the third one is from PMMA75 to PMMA100. In the first range, crystallization slowly decreases (see Section 3.2) but the addition of PMMA, which has a much higher overall tensile strength response than PVDF, strengthens the material. By comparing the tensile strength responses between the two processing techniques from 0–25 wt% PMMA, tensile strength converges while the overall tensile strength increases with PMMA content. Specifically, the percent difference between processing methods decreases from 18% to 11% for 0 wt% to 25 wt% PMMA, respectively. This convergence in the tensile strength response of IM and 3D printed samples is due to PMMA acting as an adhesive. The latter enables higher mobility and diffusion of polymer chains at lower temperatures owing to its high surface energy, which raises blends overall surface energy. Therefore, this adhesive mechanism helps to increase the bond formation of 3D-printed samples.

At PMMA50, a notable deterioration of mechanical properties can be observed. In particular, the tensile strength response of both IM and 3D printed specimens drops off due to loss of crystallization. As explained in Section 3.2, PMMA50 is the limit concentration beyond which the PVDF crystallization event is lost. The PMMA concentration is too high to allow for PVDF crystallization, which increases the tensile strength in the polymer. More specifically, PMMA chains impede PVDF crystallization due to the high glass-transition temperature (*T_g_*) of PMMA, which causes PVDF chains to not have enough space to move and form crystals. This loss of the crystallization impact on mechanical strength is responsible for the drop in PMMA50 tensile strength value. Note that the latter is not a bond formation issue because the drop in mechanical properties occurs for both processed samples. Thus, what changes as a function of the composition (i.e., it is not processing technique dependent) is the fact that PMMA50 is no longer able to crystallize and, hence, it has lost part of its load-bearing ability.

From PMMA75 to PMMA100, the increase in PMMA content raises tensile strength simply because the overall PMMA tensile strength response is much higher than that of PVDF. In this way, the mechanism for improving this mechanical property is simply the fact that PMMA itself has a higher tensile strength than PVDF. Within this range, blends tensile strength responses are approaching the PMMA response. However, the difference between the tensile strength response of IM and 3D printed specimens is noticeably greater than it is for lower PMMA concentrations. These differences are greater because blends glass-transition temperatures have notably increased close to the value of virgin PMMA *T_g_* (i.e., 118 °C). The relatively high glass-transition of PMMA creates diffusion limitations as the PMMA network becomes rigid below *T_g_*. These limitations hinder domain evolution at the printed filament boundary, effectively seizing polymer welds at high temperatures, thus limiting the degree of diffusion from layer to layer.

Concerning the elongation at break, a similar but decreasing trend can be observed with the addition of PMMA weight ratios. Starting from PMMA0, ductility decreases with increasing PMMA concentration. As seen in the tensile strength trend, a significant drop in elongation is identified at PMMA50. This drop in ductility can be explained by the fact that PMMA50 samples can only withstand a much lower load for a shorter time than the other bends due to (1) the loss in PVDF crystallization and (2) PVDF chains disrupting the PMMA regions, effectively reducing the load-bearing potential. Hence, their load-bearing potential drops. From PMMA75 to PMMA100, ductility follows the same decreasing trend that it had before PMMA50 until reaching PMMA100. High PMMA concentrations exhibit brittle behavior in response to applied loads mainly because they have high M_w_ and were tested below their *T_g_* (see next section).

Finally, Young’s modulus trend matches tensile strength as it slowly increases proportionally to the amount of PMMA. In addition, the same drop in property can be observed at PMMA50. This intermediate composition does not have the crystallization impacts that help to increase load-bearing potential and does not have enough PMMA content to approach the modulus of the high PMMA concentration specimens. While the processing method strongly impacts specimen tensile strength and elongation, Young’s modulus exhibits very small changes (i.e., an average of 4% difference) between the two types of production. The latter can be explained because Young’s modulus is a material-dependent property, whereas the former properties depend more on production. This material dependence can be visualized in that even if the geometry of the sample were to be changed, Young’s modulus would not change. In contrast, the first two properties, which are more process-dependent, are very sensitive to voids and weld quality (i.e., stress concentrations and interlayer adhesion effects). As explained later in the GPC and FTIR analysis results, the internal properties (i.e., M_w_ and crystalline phases) of the material have not changed and remain fairly constant between processed specimens.

### 3.4. Attenuated Total Reflectance Spectroscopy (ATR-FTIR)

To study the phase domains of PVDF/PMMA mixtures (i.e., PVDF crystalline phases), processed compositions underwent ATR-FTIR analysis. The latter measures the infrared spectrum of absorption of the sample over a range of wavelengths (see Appendix A). In this way, absorbance spectra allow the determination of the morphology of the polymer blend by observing which bonds and phases are present in the material. Therefore, PVDF characteristic peaks, as well as PMMA frequency bands of interest, need to be examined.

Appendix A shows the infrared spectra of the different PVDF/PMMA processed blends from 600 to 1800 cm^−1^. As previously mentioned, PVDF can crystalize into five different crystalline phases (polymorphs): α, β, γ, δ and ε [14,15]. The first three phases (α, β, ) are the most commonly studied, with the α-phase being the most thermodynamically preferred [16]. Only the representative peaks of α- (614, 762, 795, 855, and 976 cm^−1^) and β- (840 and 1279 cm^−1^ essentially) phases are present in the spectrum of blends. More specifically, only concentrations up to 25% in PMMA content display these peaks. Beyond this limit concentration, α and β peaks diminish in intensity or vanish, which agrees with the crystallinity results discussed in the previous section. Moreover, pure PMMA displays at 1720 cm^−1^ the stretching frequency of its carbonyl group (C = O). However, for blends containing both PVDF and PMMA, this absorption band is shifted to higher wavelengths (1727–1729 cm^−1^) due to the interaction between the carbonyl groups of PMMA and the CH_2_ groups of PVDF, which indicates the formation of blends [29,31]. It can also be observed that blends containing high PMMA content display at 990 cm^−1^ the bending frequency of CH_3_-O [32].

Investigation of the crystalline domains present in each mixture provides information about the diffusion environment. For instance, if PMMA interacts strongly with PVDF through its carbonyl group, then the β-phase should be more dominant than the α-phase. Consequently, the higher the β-phase fraction, the greater the intermolecular interaction between PVDF and PMMA, which will subsequently negatively affect diffusion. Alternatively, if it is mainly the α-phase that is present within the blends, there will be fewer PVDF-PMMA interactions and thus a higher possibility of diffusion across the polymer interfaces. Thus, the β-phase content of blends containing only α and β polymorphs can be calculated using the following Equation (5), where *F(β)* represents the amount of β-phase; *A_α_* and *A_β_* are the absorbance intensities for α- and β-phase at 762 and 840 cm^−1^, respectively; *K_α_* and *K_β_* are the absorption coefficients at the respective wavenumbers (i.e., *K_α_* = 6.1·10^4^ cm^2^/mol and *K_β_* = 7.7·10^4^ cm^2^/mol) [33,34]:(5)$$F\left(\beta \right)=\frac{A_{\beta }}{\frac{K_{\beta }}{K_{\alpha }}A_{\alpha }+A_{\beta }}\times 100$$

Surprisingly, the results presented in Figure 4 show a decrease in the relative fraction of β-phase for the processed samples as the PMMA content increases, which runs counter to the findings of the existing literature [19]. Only the transition from PMMA15 to PMMA25 of the injection-molded sample shows a slight increase in the β-phase content. Still, these trends are consistent with the previously reviewed results. Hence, the increase in PMMA content not only worsens PVDF crystallization but also hinders nucleation of the β-phase PVDF polymorph. The total amount of β-phase in absolute terms can be calculated by combining these results with the DSC crystallinity calculations.

It can be noted that injection-molded specimens present higher absorbance intensities than 3D-printed specimens, which can be explained by the sample transparency. The 3D printed sample is opaquer due to its interfaces, which refract light more efficiently than the transparent IM sample. Thus, when the machine detects the reflected infrared light from the multilayer sample, this return light is more scattered, and the detector cannot capture it as well as with the injection-molded sample. Even so, this issue does not affect the determination of the β-phase content since Equation (5) works with relative amounts of the same spectrum. It is worth mentioning that ATR-FTIR spectroscopy was used instead of FTIR transmission spectroscopy because samples were very opaque, and light could not pass through properly. Reflectance is a superficial measurement, while transmittance is a bulk measurement.

### 3.5. Gel Permeation Chromatography (GPC)

GPC was run to study the processing effects of injection molding, 3D printing, and twin-screw extrusion on the molecular weight M_w_ and polydispersity of PMMA and PVDF. GPC curves are displayed in Appendix A, while its number and weight averaged molecular weights and dispersity values are shown in Table 3 and Table 4. From GPC curves it can be observed that the molecular weight (M_w_) is higher in PMMA than PVDF since its narrow peak occurs early in the range of 10–14 min whereas the PVDF peak takes place later between 24 and 30 min. Moreover, PVDF exhibits a lower intensity peak around 40 min, which suggests that its low M_w_ chains act as a plasticizer and soften it. Therefore, PVDF chains require less energy (i.e., temperature) to experience motion than PMMA chains. In this way, PVDF chains can diffuse into PMMA and reduce large chain interactions with one another, improving blend plasticity.

Results show that both processed thermoplastics did not experience degradation because they maintain a fairly constant molecular weight distribution throughout the eluded time for each production technique. This can also be inferred from the data presented in Table 3 and Table 4, as the number and weight average molecular weights values are within an acceptable range of variation for PMMA and PVDF. Therefore, for the tensile test performed, differences between the responses of IM and 3D printed specimens can be argued from a diffusion point of view (i.e., no change in material properties). Similarly, it can be verified that the thermal analysis was carried out in the appropriate temperature range because there is no change in thermal behavior due to the decomposition of polymer chains. It is important to note that degradation would imply that polymer chains are being cut (chain scissioning) during blends production and hence there would be a clear difference in the molecular weight distribution of the different samples.

## 4. Conclusions

In this study, PVDF/PMMA dog-bone specimens were fabricated via FDM and injection molding. Specimen mechanical and thermal properties were evaluated by tensile testing and DSC, respectively. Tensile testing results proved PMMA to be the strongest material with the highest Young’s modulus. In contrast, results revealed that PVDF has the highest elongation at break. When PMMA content ratios increased for each mixing ratio, the strength increased as well as Young’s Modulus, whereas elongation at break decreased. However, PMMA50 was demonstrated to differ from this trend to be the weakest and most brittle composition due to the loss of crystallization. These results help to assess the impact of PMMA and the FDM process on PVDF/PMMA blends behavior and processability.

DSC results revealed that the increase in PMMA content disrupted PVDF crystalline domains. The results suggested that PMMA50 is the limiting concentration beyond which the PVDF crystallization event is lost. Furthermore, PVDF crystalline phases were examined using the ATR-FTIR technique. Results revealed that for high concentrations of PVDF, an increase in PMMA content hindered nucleation of the β-phase PVDF polymorph. GPC results showed that both processed polymers did not experience degradation as specimens examined maintained a fairly constant molecular weight distribution throughout the eluted time. Finally, this work provides insight into the multiscale mechanical behavior of PVDF/PMMA structures processed through FDM from which interesting technical advances can be made in different fields by understanding these fundamental behaviors to the point of intentional manipulation.

## Figures and Tables

**Figure 1 materials-15-08062-f001:**
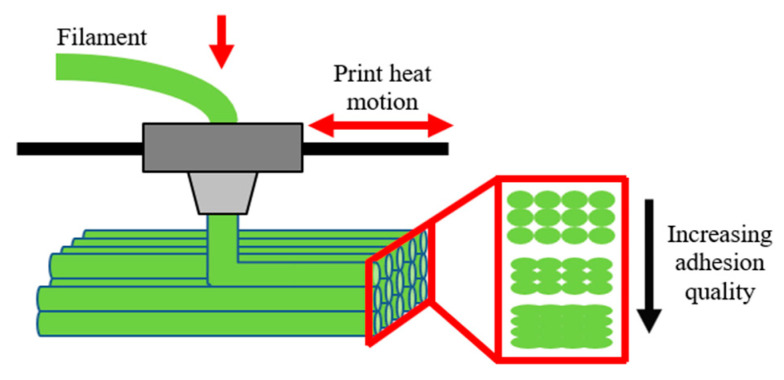
Schematic illustration of the adhesion quality importance during FDM.

**Figure 2 materials-15-08062-f002:**
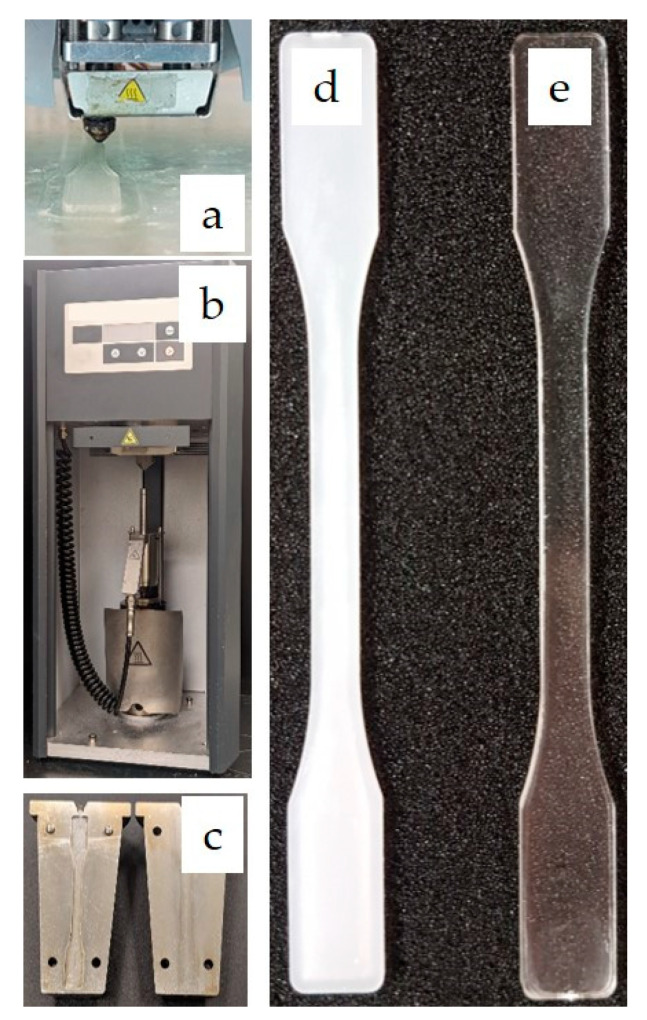
Image of (**a**) the FDM process, (**b**) the injection molding equipment, (**c**) mold geometry, (**d**) sample dogbone (PMMA0) made via FDM, and (**e**) sample dogbone (PMMA100) made via injection molding.

**Figure 3 materials-15-08062-f003:**
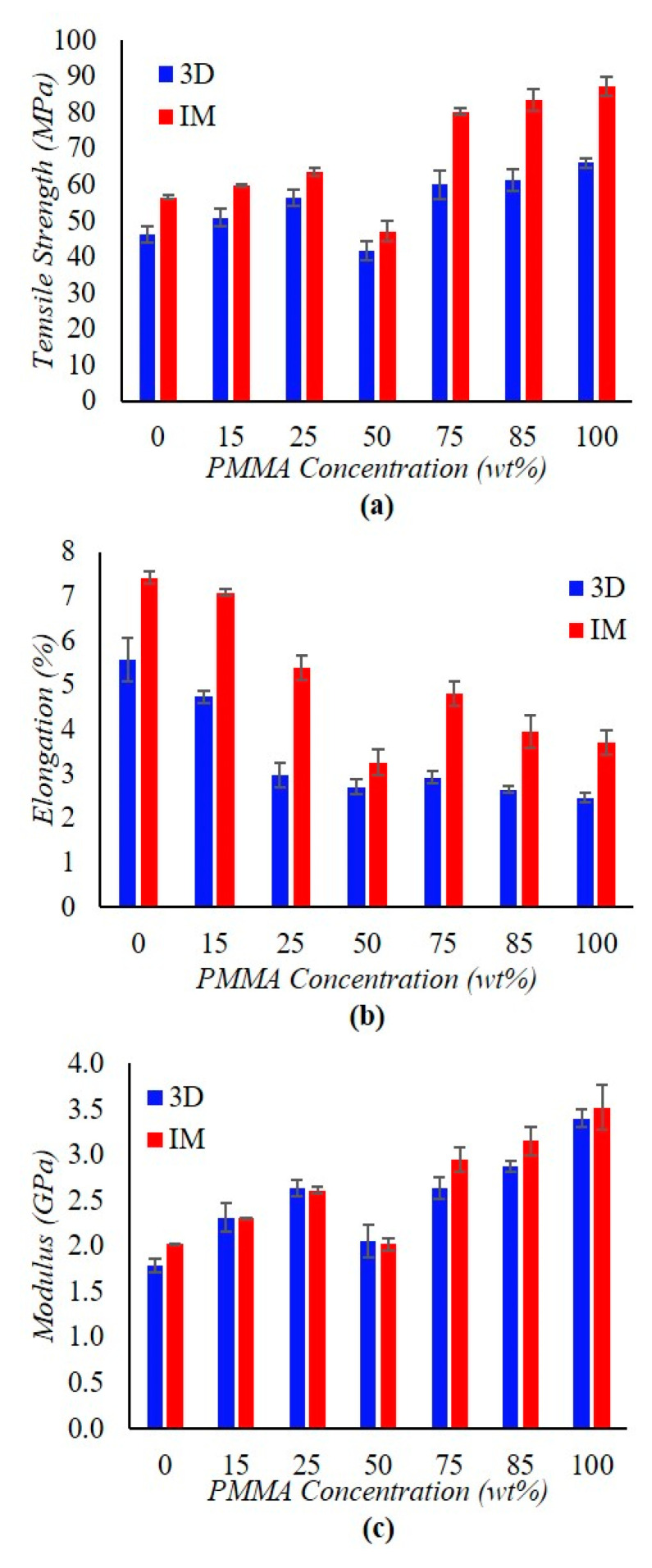
Mechanical properties of injection molded (IM) and 3D printed (3D) PVDF/PMMA dog-bones as a function of PMMA content: (**a**) tensile strength, (**b**) elongation at break, and (**c**) Young’s modulus. In each panel, error bars represent the standard error of the mean values of each mechanical property.

**Figure 4 materials-15-08062-f004:**
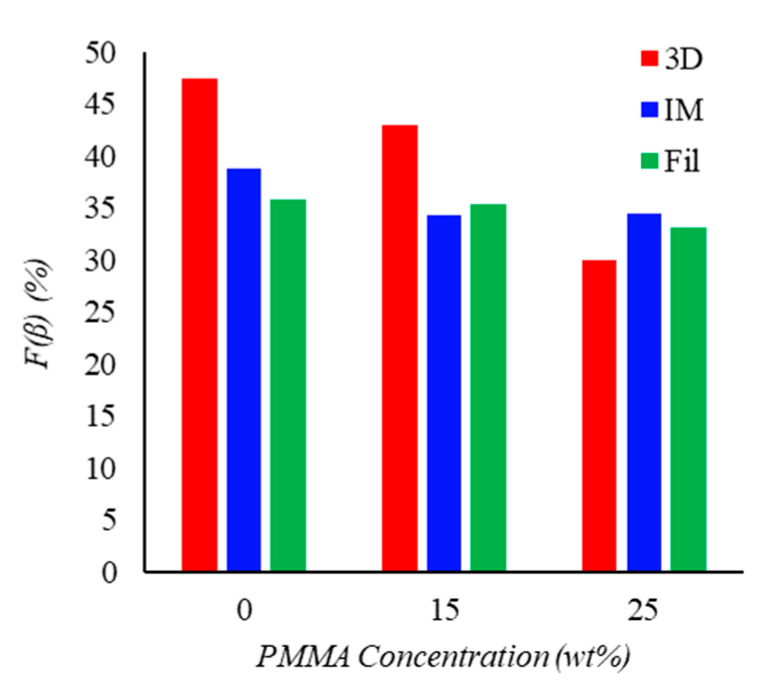
β-phase percent ratio in PVDF/PMMA processed blends as a function of PMMA content.

**Table 1 materials-15-08062-t001:** Apparent viscosity calculations from Equation (1) using Δ*p* values measured during compounding.

Sample	Apparent Viscosity [kPa·s]
PMMA0	60.37 ± 0.51
PMMA15	65.05 ± 0.38
PMMA25	74.75 ± 0.44
PMMA50	147.08 ± 6.97
PMMA75	223.47 ± 2.22
PMMA 85	235.49 ± 3.75
PMMA 100	247.16 ± 5.35

**Table 2 materials-15-08062-t002:** Glass transition (T_g_), melting temperature (T_m_), melting enthalpy (H_m_), and crystallization (X_c_) for the first DSC heating cycle. Note: --indicates no measured data.

Filament	3D Printed				IM				Filament			
	*T_g_* (°C)	*T_m_* (°C)	*H_m_* (J/g)	*X_c_*	*T_g_* (°C)	*T_m_* (°C)	*H_m_* (J/g)	*X_c_*	*T_g_* (°C)	*T_m_* (°C)	*H_m_* (J/g)	*X_c_*
*PMMA0*	*--*	164	45.0	43%	--	166	37.8	36%	--	161	38.4	37%
*PMMA15*	54	157	35.5	40%	58	162	30.6	34%	57	160	30.2	34%
*PMMA25*	57	160	28.8	37%	60	164	32.1	41%	61	157	21.0	27%
*PMMA50*	61	146	16.5	32%	62	151	11.9	23%	63	156	0.6	1%
*PMMA75*	88	--	--	0%	86	--	--	0%	99	--	--	0%
*PMMA85*	98	--	--	0%	101	--	--	0%	110	--	--	0%
*PMMA100*	111	--	--	0%	106	--	--	0%	112	--	--	0%

**Table 3 materials-15-08062-t003:** Number ($\bar{M_{n}}$ ) and weight ($\bar{M_{w}}$ ) averaged molecular weights and dispersity index (Ð) for pellet and processed PMMA samples.

Sample	$\bar{M_{n}}$ (g/mol)	$\bar{M_{w}}$ (g/mol)	Dispersity $\left(Ð=\bar{M_{w}}/\bar{M_{n}}\right)$
Pellet	8.95 × 10^4^ (±2.21%)	1.41 × 10^5^ (±1.33%)	1.57 (±2.58%)
Filament	9.08 × 10^4^ (±1.37%)	1.46 × 10^5^ (±1.08%)	1.61 (±1.74%)
Injection Molded	8.29 × 10^4^ (±4.76%)	1.33 × 10^5^ (±2.12%)	1.59 (±5.21%)
3D Printed	8.62 × 10^4^ (±3.71%)	1.37 × 10^5^ (±1.73%)	1.62 (±4.09%)

**Table 4 materials-15-08062-t004:** Number ($\bar{M_{n}}$ ) and weight ($\bar{M_{w}}$ ) averaged molecular weights and dispersity index (Ð) for pellet and processed PVDF samples.

Sample	$\bar{M_{n}}$ (g/mol)	$\bar{M_{w}}$ (g/mol)	Dispersity $\left(Ð=\bar{M_{w}}/\bar{M_{n}}\right)$
Pellet	4.57 × 10^4^ (±7.11%)	8.27 × 10^4^ (±1.19%)	1.81 (±7.21%)
Filament	4.41 × 10^4^ (±9.72%)	8.24 × 10^4^ (±1.77%)	1.87 (±9.88%)
Injection Molded	4.13 × 10^4^ (±8.38%)	8.25 × 10^4^ (±1.31%)	1.99 (±8.49%)
3D Printed	4.09 × 10^4^ (±8.66%)	8.02 × 10^4^ (±1.27%)	1.96 (±8.75%)

## Data Availability

Not applicable.

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
