# Peer review of "Exploiting Partial Solubility in Partially Fluorinated Thermoplastic Blends to Improve Adhesion during Fused Deposition Modeling"

_materials, 2022, doi:10.3390/ma15228062_

Round 1

Reviewer 1 Report

It is a well-written manuscript. It was very interesting to read the manuscript. The text is written quite clearly. The main merit of the manuscript is the Results and Conclusion section, in this section the authors provided necessary and detailed information about the adhesion mechanism of diffusion and entanglement of PVDF/PMMA composite, which did not raise serious questions. Nevertheless, some comments and clarifying questions are given below: 

- Perhaps in the Materials and Methods section, photos of samples obtained by 3D printing and Injection Molding should be added.

- Were the SEM sections of the specimens analyzed after tensile testing?

- What is the preferred percentage of PMMA in the composite according to the authors (based on all the studies conducted)? Perhaps this should be added to the "Conclusion".

Overall, this paper left a positive impression and may be of interest to academic readers of the journal. 

Author Response

It is a well-written manuscript. It was very interesting to read the manuscript. The text is written quite clearly. The main merit of the manuscript is the Results and Conclusion section, in this section the authors provided necessary and detailed information about the adhesion mechanism of diffusion and entanglement of PVDF/PMMA composite, which did not raise serious questions. Nevertheless, some comments and clarifying questions are given below: 

Q1: Perhaps in the Materials and Methods section, photos of samples obtained by 3D printing and Injection Molding should be added.

A1: These images have been added as requested within Figure 2.

Q2: Were the SEM sections of the specimens analyzed after tensile testing?

A2: SEM was not performed on these specimen.

Q3: What is the preferred percentage of PMMA in the composite according to the authors (based on all the studies conducted)? Perhaps this should be added to the "Conclusion".

A3: Preference is user/application specific. Disrupting crystallinity at the inter-layer interface promotes better diffusion, but if the application requires high crystallinity (e.g., piezoelectric components), this would not be desired.

Overall, this paper left a positive impression and may be of interest to academic readers of the journal. 

Author Comment: Thank you for your review!

Reviewer 2 Report

The work presented in the paper entitled "Exploiting Partial Solubility in Partially Fluorinated Thermo- 2 plastic Blends to Improve Adhesion During Fused Deposition 3 Modeling" is novel. But the following comments need to be addressed

1) Line 72 - "-40 °C and a melting point around 170 °C" Provide appropriate reference.

2) Line 113-121 - "All technical information provides in the tabular form."

3) Line 163 - Why authors cooled samples to 30 °C? Any specific reason?

4) Provide the image of tensile testing.

5) What is the outcome of the tensile test? Provide the graphical information.

6) Line-250  Differential scanning calorimetry (DSC) provides the graph of heat flow Vs. Temperature.

Author Response

The work presented in the paper entitled "Exploiting Partial Solubility in Partially Fluorinated Thermo- 2 plastic Blends to Improve Adhesion During Fused Deposition 3 Modeling" is novel. But the following comments need to be addressed

Q1) Line 72 - "-40 °C and a melting point around 170 °C" Provide appropriate reference.

A1: A reference has been added for these values.

Q2) Line 113-121 - "All technical information provides in the tabular form."

A2: We did not find this quoted text in the manuscript and the reviewer does not offer instructions with the text.

Q3) Line 163 - Why authors cooled samples to 30 °C? Any specific reason?

A3: This is a standard practice when operating an auto-sampling DSC so that the initial testing temperature is consistent for all samples.

Q4) Provide the image of tensile testing.

A4: We do not have images of tensile testing in action. However, we have provided the instrument’s model number among other specifications to ensure repeatability alongside stress-strain plots in the supporting information.

Q5) What is the outcome of the tensile test? Provide the graphical information.

A5: We have added a stress-strain plots as Figures S1-S14 in the supporting information.

Q6) Line-250  Differential scanning calorimetry (DSC) provides the graph of heat flow Vs. Temperature.

A6: These plots are available in the supporting information.

Reviewer 3 Report

The paper entitled ‘Exploiting Partial Solubility in Partially Fluorinated Thermoplastic Blends to Improve Adhesion During Fused Deposition Modeling’ is focused on PVDF/PMMA blends. Obtained results are well presented. Below you can find comments.

Results and discussion:

1.       Please explain why the ATR-FTIR technique was used for the polymorphism study? Add more literature examples of the usage of ATR-FTIR for this purpose.

2.       Please add to the text the information about the glass temperature of PVDF.

3.       The table or list with the name of the samples should be added. What is the composition of the sample e.g. PMMA15?? 15 means concentration of the PMMA??

4.       Did you perform the GPC measurements for blends??   

Conclusion

1.       The conclusion is too long and should be rewritten. The conclusion should highlight the most important results and main findings.

Author Response

The paper entitled ‘Exploiting Partial Solubility in Partially Fluorinated Thermoplastic Blends to Improve Adhesion During Fused Deposition Modeling’ is focused on PVDF/PMMA blends. Obtained results are well presented. Below you can find comments.

Results and discussion:

Q1.       Please explain why the ATR-FTIR technique was used for the polymorphism study? Add more literature examples of the usage of ATR-FTIR for this purpose.

A1: This is common practice for studying polymer polymorphism and is explained in the text between lines 413 and 427. We have included 8 references by which polymorphism is identified via ATR-FTIR throughout this section.

Q2.       Please add to the text the information about the glass temperature of PVDF.

A2: This is present on line 72.

Q3.       The table or list with the name of the samples should be added. What is the composition of the sample e.g. PMMA15?? 15 means concentration of the PMMA??

A3: Explanation of the nomenclature is present between lines 97 and 101.

Q4.       Did you perform the GPC measurements for blends??   

A4: We did not perform GPC measurements on the blends.

Reviewer 4 Report

Výsledky překladu

  The article deals with the study of the properties of PVDF/PMMA binary blends. The properties of the product (test specimens) from a PVDF/PMMA blend with different concentrations of PMMA produced by 3D printing (FDM method) and injection molding are assessed. The Abstract and Introduction are written concisely and comprehensibly, the literature is cited in the article with the exception of publications 12, 13, 30 and 31, which I could not find in the text. The materials and methods used are described very well and to a sufficient extent. I recommend using only SI units (see line 181 ch. 2.5.3 Tensile testing). The article is prepared carefully, clearly and comprehensibly and brings interesting information. The results are presented clearly and comprehensibly, the conclusion sufficiently summarizes the obtained results of the study. The relatively significant change in mechanical properties found during tensile tests for a concentration of 50% PMMA in the blend appears to be very interesting. Due to the width of the interval (25-75%PMMA), it is difficult to describe this change with sufficient explanatory power. A deeper understanding of these changes would benefit from narrowing the interval between PMMA doses (e.g. 35, 35, 50, 65 and 75% PMMA). There is only one measured value between the doses of 25 and 75% PMMA. Why were these PMMA intervals/distances chosen?

Author Response

 The article deals with the study of the properties of PVDF/PMMA binary blends. The properties of the product (test specimens) from a PVDF/PMMA blend with different concentrations of PMMA produced by 3D printing (FDM method) and injection molding are assessed.

Q1: The Abstract and Introduction are written concisely and comprehensibly, the literature is cited in the article with the exception of publications 12, 13, 30 and 31, which I could not find in the text.

A1: Thank you for caching this error. The references have been updated to remove extra citations.

Q2: The materials and methods used are described very well and to a sufficient extent. I recommend using only SI units (see line 181 ch. 2.5.3 Tensile testing).  

A2: Thank you for catching this. 30 ft.lb has been changed to 40 N.m.

The article is prepared carefully, clearly and comprehensibly and brings interesting information. The results are presented clearly and comprehensibly, the conclusion sufficiently summarizes the obtained results of the study. The relatively significant change in mechanical properties found during tensile tests for a concentration of 50% PMMA in the blend appears to be very interesting. Due to the width of the interval (25-75%PMMA), it is difficult to describe this change with sufficient explanatory power. A deeper understanding of these changes would benefit from narrowing the interval between PMMA doses (e.g. 35, 35, 50, 65 and 75% PMMA).

Q3: There is only one measured value between the doses of 25 and 75% PMMA. Why were these PMMA intervals/distances chosen?

A3: As we designed this study, we wanted to see if we could minimize the amount of PMMA necessary for printability. Most of the applications in which we are interested rely on PVDF for functionality. We also wanted to probe solubility limits, which occur in high PMMA concentrations. As a result, we investigated the two extremes more detailed than the moderate concentrations. In future work, we can certainly expand on this region, but the interest is not as strong since the most detrimental mechanical performance occurred at the intermediate compositions.

Round 2

Reviewer 3 Report

Manuscript can be published at the present form.